# Exploring the Global Research Trends of Supply Chain Management of Construction Projects Based on a Bibliometric Analysis: Current Status and Future Prospects

**Shiping Wen** [1]**, Handong Tang** [2,]*****, Fei Ying** [3] **and Guangdong Wu** [2]

[1] School of Civil Engineering and Architecture, Chongqing University of Science and Technology, Chongqing 401331, China
[2] School of Public Policy and Administration, Chongqing University, Chongqing 400044, China
[3] Department of Built Envinroment Engineering, Auckland University of Technology, Auckland 1010, New Zealand
***** Correspondence: 20220101010@stu.cqu.edu.cn

**Abstract:** With the increasing scale and complexity of construction projects, a considerably growing number of studies have focused on managing supply chain scientifically to improve performance. To explore this field in depth, this paper uses the Bibliometrix R encapsulation tool to conduct a bibliometric analysis and visually display key findings on construction project supply chain management from 1998 to 2021. By using a series of indexes in econometric analysis, this paper introduces an overview of construction project supply chain research. Further, the current situation, historical evolution, and development of this field are explored using the content analysis of keywords. The results suggest that: (1) the number of publications in the field of the supply chain management of construction projects has increased over time and entered a period of rapid growth after 2015. During this period, articles related to 'sustainability', 'prefabricated housing', 'reverse logistics', and 'lean management' were widely cited by scholars, but the most frequently cited was 'partnership'. (2) The keywords in the field of construction project supply chain management can be divided into 'simulation research', 'sustainable research', 'method research', 'cooperation and integration', 'whole life cycle,' 'trust and communication', and so on. The keywords 'Radio Frequency Identification' (RFID), 'waste management', 'case study', and 'social responsibility' are the current research hotspots. (3) The theme development of construction project supply chain management can be divided into supply chain integration and management, supply chain process design and optimization, the application of advanced technology, and sustainable supply chain management. This paper summarizes the main discoveries and developments in construction project supply chain management.

**Keywords:** construction project; supply chain management; bibliometric research; content analysis; bibliometrix

## 1. Introduction

In the 1990s, the manufacturing concept of the supply chain (SC) was introduced to the construction field [1]. With the increasing scale and complexity of construction projects, the construction supply chain (CSC) has developed from a simple linear chain structure to a complex network structure. An increasing number of construction-related institutions began to manage their SC scientifically to improve performance [2,3]. Today, construction companies prefer to compete in the form of an SC rather than as a single company. A large number of practices have shown that effective supply chain management (SCM) can reduce the total cost of a project by 5% and 15% per year [4]. In a construction project, the SC is composed of organizational structures for various project processes, which can be regarded as a network composed of information flow, logistics, capital flow, and service or product relationships among stakeholders [5]. SCM theories guide project managers to

achieve higher management efficiency and add value to all stakeholders in the SC [6]. At the same time, SCM can productively enhance knowledge exchange among partners in the planning, design, construction, and maintenance stages [5]. Therefore, in recent years, construction supply chain management (CSCM) has attracted attention from academics, various construction firms, research institutions, governmental management departments, and the public.

In terms of structure and function, the CSC is a temporary SC, unique and customized. CSCs are thus unstable, fragmented, and cannot be completely repeated [7]. Despite numerous studies in the field of CSCM, challenges associated with information exchange [8], the lag of knowledge transfer [9], inefficient cooperation [10], the conflicts of interest among stakeholders remain [11]. Therefore, the research gaps and cutting-edge research directions in CSCM research need to be identified by sorting out the existing CSCM research results with a holistic understanding of the current state of research in the field.

A literature review of the CSCM field is an opportunity for researchers to explore the CSCM field from different perspectives. This is conducive to showing researchers the real research status in the field, helping researchers better summarize existing research and grasp future research trends. The existing research has launched a series of research reviews in the field of CSCM. From the perspective of content, scholars have different research focuses on CSCM in different periods. At the beginning of this century, researchers' research goal was to deepen their understanding of CSCM. Vrijhoef and Koskela (2000) studied CSCM through a case analysis and comparison with previous studies and introduced the four functions of SCM in the construction industry [7]. Saad and Jones et al. (2002), on the basis of a literature review and survey from 1960 to 2000, proposed that construction industry practitioners need a better conceptual understanding of CSCM and a more systematic implementation method [12]. Azambuja and Brien (2009) found that the most important factors for different components of the supply chain are: CSC configuration, partner selection, information system planning, and risk management [13]. Cheng and Law et al. (2010) summarized the overall structure and characteristics of a service-oriented architecture supply chain integration framework through a literature analysis. In recent years, with the increasing diversity of engineering projects and the continuous enrichment of management means, researchers have paid attention to the integrated management of CSCM as an effective way and tool to achieve project management through research on supply chains with different attributes [14]. Shi and Ding (2016) analyzed the literature on the supply chain management of mobile Internet-based construction projects from 1998 to 2015 and analyzed the current situation and the possibility of future research from the perspective of Internet management [15]. Le and Elmughrabi (2020) reviewed the literature on the supply chain management of construction projects, identified the current research focus, and discussed the future development direction of construction supply chain management research [16]. Liu and Dong (2020) discussed the research focus of the supply chain management of prefabricated construction projects by analyzing the literature on supply chain management from 2001 to 2018 [17]. In literature review research, qualitative research can predict the development and trend of future research based on the analysis, summary, collation, and review of existing research [7,12,14,18]. However, qualitative methods can only be based on the researchers' induction and summary of the field, so scholars have strong subjectivity in the selection of documents. This will mean that in many research results, all of the articles in the research field cannot be analyzed. Moreover, this kind of literature mainly takes a country or region as the research area, lacks an international perspective, and cannot accurately grasp the global situation of construction project supply chain management.

As a quantitative research method, bibliometrics takes the external characteristics of documents such as the distribution structure, quantitative relationship, and change law of documents as the research object and conducts a detailed analysis of documents in the field [19]. At present, scholars have conducted quantitative literature analyses on the supply chain management of construction projects [15–17]. However, the current quantitative research and analyses mainly focus on the law of changes in literature publication year by

year, the clustering analysis of article keywords, and the co-occurrence analysis of authors, research institutions, and countries. Research on citation analysis, cited analysis, and theme evolution are trends. In view of this, this paper uses the R language software package as an analysis tool for bibliometric analysis to conduct a comprehensive quantitative analysis and evaluation of the literature in the field of construction project supply chain management in the core database of SciDev.Net from 1998 to 2021. This will describe the overall situation of construction project supply chain management in detail and comprehensively, from a global perspective, establish the overall knowledge framework of construction project supply chain management research and review the development trends in this field. This will not only help to objectively reveal the current research situation of the supply chain management of construction projects but also provide a scientific reference for the future research direction of the supply chain management of construction projects, and it will also have certain reference significance for information science research in other fields. Therefore, the research objectives of this paper are as follows:

I       What is the status of research in the field of CSCM?
II      What are the main research directions of CSCM?
III     How did the theme of CSCM evolve?
IV      What is the potential research direction of CSCM in the future?

This paper expounds on the basic rules of CSCM from the aspects of annual literature, research leadership (e.g., country, author, and journal), research hotspots, and research topics. This paper analyzes the current research situation and historical dynamic evolution of CSCM from the perspectives of historical citation and discipline evolution. The approach provides references and suggestions for future CSCM research. The rest of the paper is arranged as follows: Section 2 describes the research methods and the acquisition of research data. Then, Section 3 first summarizes and analyzes the current situation of the literature, and then measures the literature through keyword analysis to effectively achieve the research objectives proposed in this study. After that, the future research trends are discussed in Section 4. Section 5 provides suggestions, and Section 6 summarizes the findings and further points out the future research needs.

## 2. Research Methods

This paper thus uses a comprehensive bibliometric method to explore the field of CSCM by combining econometric analysis with content analysis. In this step, the authors identified the candidate keyword(s), selected the most appropriate keywords, and conducted the database search, which is explained more in Section 2.2 of the manuscript.

### 2.1. Methods

The standard bibliometric analysis process includes five steps: research design, data collection, data analysis, data visualization, and data interpretation [20]. Currently, several software packages are available for the measurement of document information based on the R language. For example, Citan can be used to calculate various literature metrics, such as the h-index and l-index. However, compared with bibliometric analysis, it lacks the functions of co-citation analysis [21]. Similarly, Science Text lacks data import and transformation modules [22]. This paper thus uses a comprehensive bibliometric method to explore the field of CSCM by combining econometric analysis with content analysis. Figure 1 illustrates the research framework applied in the study.

### 2.2. Data Sources

The first step of this study was the selection of a database to collect high-quality data for bibliometric analysis. Using previous bibliometric research as a reference [23], the Web of Science (WOS) was used for data collection, considering that more than 11,000 authoritative and highly influential academic journals are listed in the WOS. The second step was the extraction and filtering of the data. In previous CSCM literature reviews, different retrieval codes have been used to collect data. The search strategy of Le and Elmughrabi [16]

was adopted because it examines the most diverse patterns in the CSCM search string. The following retrieval string was used in the retrieval: TS = 'construction supply chain management', 'construction supply chain logistics', 'construction supply chain modeling', 'construction supply chain management trend', and 'supply chain management evolution'. The search was conducted in December 2022, and therefore, this paper considers the period from the earliest available date until 2022. The type of literature was limited to journal articles, because this type of publication usually provides more important CSCM research with higher quality. This search yielded a preliminary list of 2785 publications. The titles, abstracts, and keywords of these records were manually checked to delete irrelevant publications. After this data-cleansing process, 503 articles that were not related to CSCM were removed. Finally, a total of 2282 bibliographic records were obtained, which were then used as the analytical dataset for this paper.

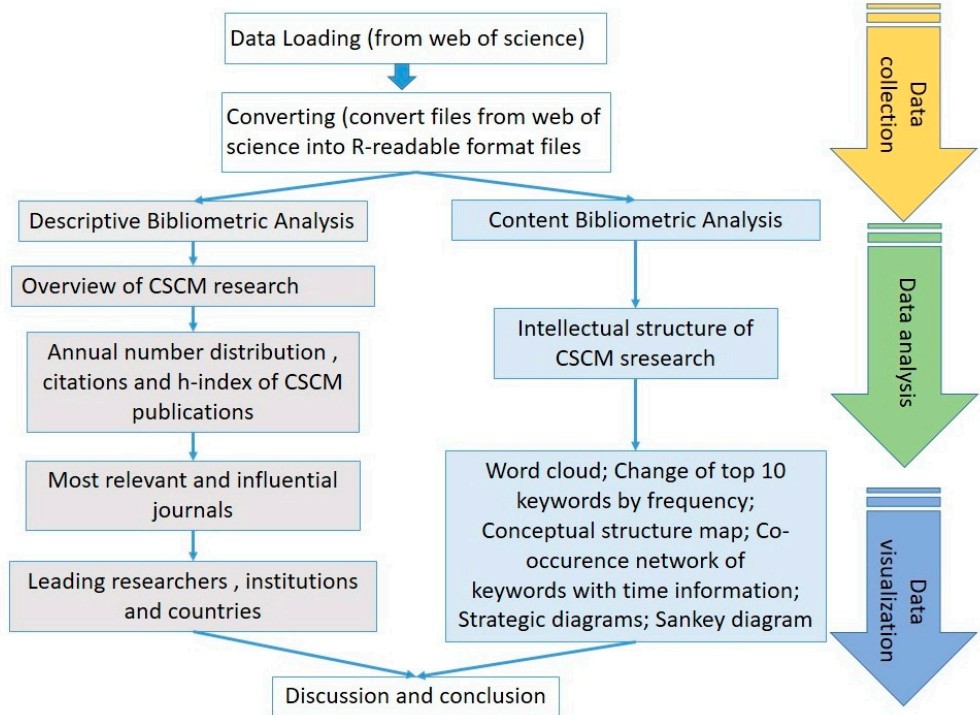

**Figure 1.** Bibliometrix and recommended science mapping workflow.

## 3. Results

Bibliometric analysis is a quantitative analysis method, which is widely used in literature review research. For example, Moosavi (2022) adopted interdisciplinary bibliometric analysis [24]. This paper firstly analyzes the annual number of publications to understand the overall development of the CSCM research field. In the second step, the country distribution of publications is analyzed to understand the development of CSCM in different countries. In the third step, the main authors in the field of CSCM are analyzed to understand the hot research fields of CSCM. Moreover, this paper analyzes the key words and puts forward suggestions for future research directions.

### 3.1. Overview of Construction Supply Chain Management Research

3.1.1. Annual Distribution and Citations of Construction Supply Chain Management Publications

Development and evolution analysis can track a target year by year according to a time series, or the target can be divided into different stages. The annual distribution of the number of published documents reflects the overall situation and research trend. The research trend indicates the overall characteristics by describing different stages of development, as shown in Figure 2. In general, the number of documents followed a

trend of volatile growth, at an annual growth rate of 24.76%, reaching a publishing peak in 2022. The studied publication timeframe can thus be divided into three stages: the initial phase, the volatility growth phase, and the rapid development stage. (I) The initial phase (1998–2006): Prior to 2006, the number of published papers increased slowly. Many studies on CSCM appeared in the form of policy recommendations with regard to cost management and project management. Few studies focused on CSCM by academics. (II) The volatility growth phase (2007–2014): The growth of the number of documents began to fluctuate. With updated construction standards and international competition, many construction companies found it necessary to integrate activities in the SC arena to balance manufacturing costs and product quality while ensuring an adequate delivery time. (III) The stage of rapid development (from 2015 to the present): The number of publications increased rapidly. CSCM is increasingly recognized by scholars, with further global research attention on construction project SCs.

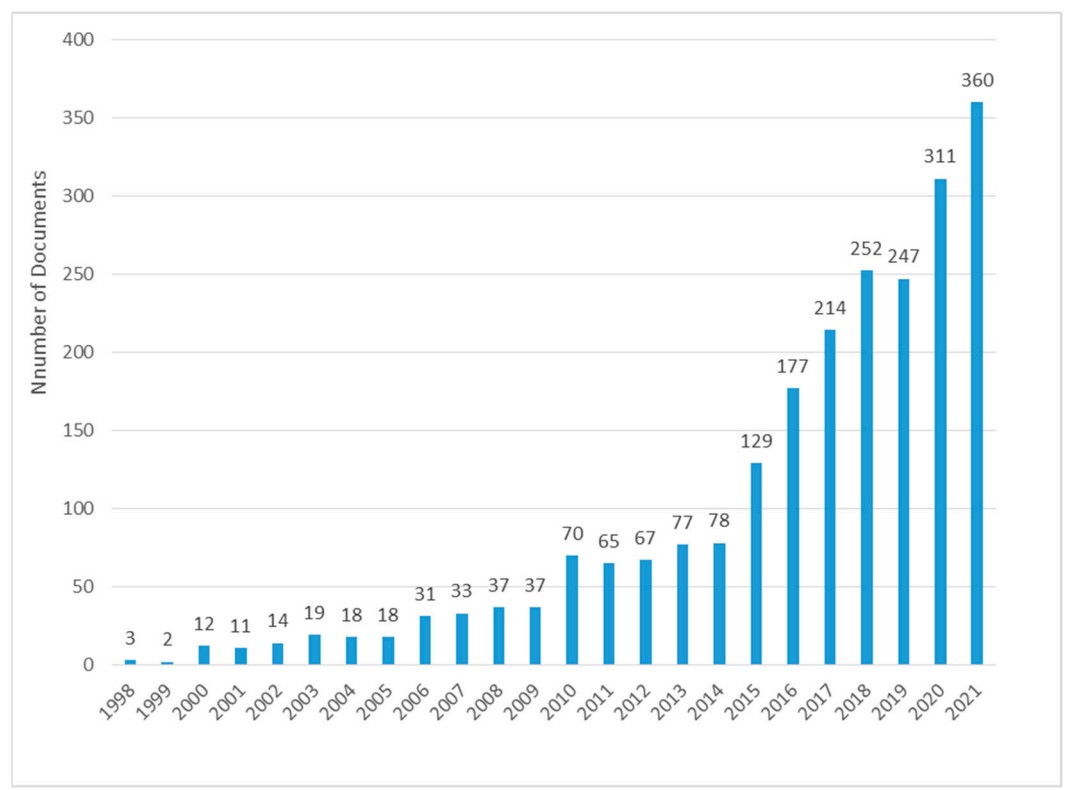

**Figure 2.** Number of publications and average citations per item in each year in the field of construction supply chain management (CSCM) publications.

### 3.1.2. Citation Analysis

A historical direct citation network was generated and analyzed visually by using the networking and component function in the literature software package. A total of 32,443 references were cited in the 2282 identified papers on CSCM, which form the basis of citations in the field of CSCM. Both the local citation score (LCS) and global citation score (GCS) were used to analyze the original data, as presented in Table 1. Figure 3 shows the historical direct citation network, where the direction arrow indicates the reference relationship between cited documents. As displayed in Figure 3, there was no blank in the literature node since 1998, indicating a high output of cited papers over the past 22 years. Several classic documents appeared from 2005 to 2015, among which, the paper with the highest LCS and GCS values was by Briscoe [25], published in *Supply Chain Management: An International Journal*. In this article, Briscoe empirically investigated the problems encountered by the British construction industry when integrating the SC, and these problems were identified through the literature on SCM. The article published by

Bankvall [26] in *Supply Chain Management: An International Journal* has also been widely recognized by scholars in the field of CSCM. This article focused on the different types of interdependences between construction project SCs and construction projects and then discusses the issues related to CSCM. The article published by Meng [27] in the *International Journal of Project Management* introduces the impact of relationship management in the supply chain on project performance. In the field of CSCM, Lrizarry [28] published a paper in *Automation in Construction*, where the annual GCS was high, but the LCS was low. This suggests that citations received by this paper are mainly from non-CSCM fields. Although citations are not high in the field of CSCM, this does not imply that this paper is of little significance in the field of CSCM. In this paper, Lrizarry discussed a method for integrating BIM and GIS into a shared system and used information technology to improve the process of CSCM integration. These top-cited papers may be the most influential in CSCM research.

**Table 1.** Local citation score (LCS) and global citation score (GCS) in construction supply chain management (CSCM) research.

| Document | DOI | Year | LCS | GCS |
|---|---|---|---|---|
| Briscoe G, 2005, *Supply chain manager* | 10.1108/13598540510612794 | 2005 | 46 | 114 |
| Bankvall L, 2010, *Supply chain manag* | 10.1108/13598541011068314 | 2010 | 28 | 68 |
| Meng XH, 2012, *Int J Proj Manag* | 10.1016/j.ijproman.2011.04.002 | 2012 | 24 | 173 |
| Cheng JCP, 2010, *Automat constr* | 10.1016/j.autcon.2009.10.003 | 2010 | 23 | 90 |
| Xue XL, 2005, *Automat constr* | 10.1016/j.autcon.2004.08.010 | 2005 | 21 | 99 |
| Behera P, 2015, *Prod Plan Control* | 10.1080/09537287.2015.1045953 | 2015 | 21 | 39 |
| Eriksson PE, 2010, *Supply chain manag* | 10.1108/13598541011068323 | 2010 | 19 | 77 |
| Khalfan MMA, 2007, *Supply chain manag* | 10.1108/13598540710826308 | 2007 | 17 | 86 |
| Irizarry J, 2013, *Automat constr* | 10.1016/j.autcon.2012.12.005 | 2013 | 17 | 146 |
| Tserng HP, 2006, *J Constr Eng M* | 10.1061/(ASCE)0733-9364(2006)132:4(393) | 2006 | 13 | 29 |

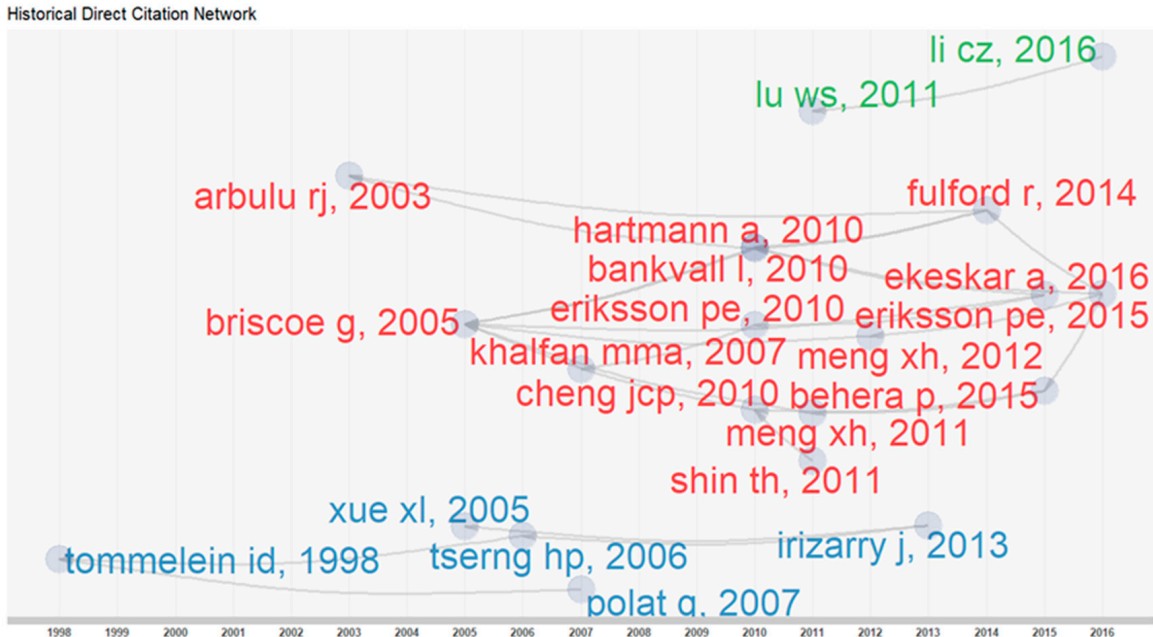

**Figure 3.** Historical direct citation network of the 20 top-cited papers in the field of CSCM from 1998 to 2021.

3.1.3. Major Country Analysis

The importance of and attention paid to CSCM research in different regions can be reflected by the publication statuses of papers in different countries. Figure 4 shows the release of articles related to the supply chain management of construction projects in

countries around the world. Between 1998 and 2020, 66 countries or regions published papers, of which, 26 countries only published one to five publications, which means that there is still a lot of space for research on supply chains of construction projects in a large number of countries. China, the United Kingdom, the United States, Australia, and Italy are the five countries with the largest number of articles published in the world. It can be seen from Figure 4 that compared with Asian countries and African countries, developed countries such as Europe and North America have published more articles, which shows that developed countries play a leading role in the field of construction project supply chains. Theoretically, the SC research of construction projects greatly impacts developing countries that are in a rapid process of urbanization; however, from the perspective of academic research, developed countries are more prominent. This is related to many factors. Among these, the level of science and technology is relatively low, and a shortage of funds is apparent. This leads to insufficient investment in the development of urbanization by a number of countries, which cannot support more academic research. Furthermore, many developing countries lack technology, capital, and talent pool; therefore, it is difficult for these to play an important role in the world.

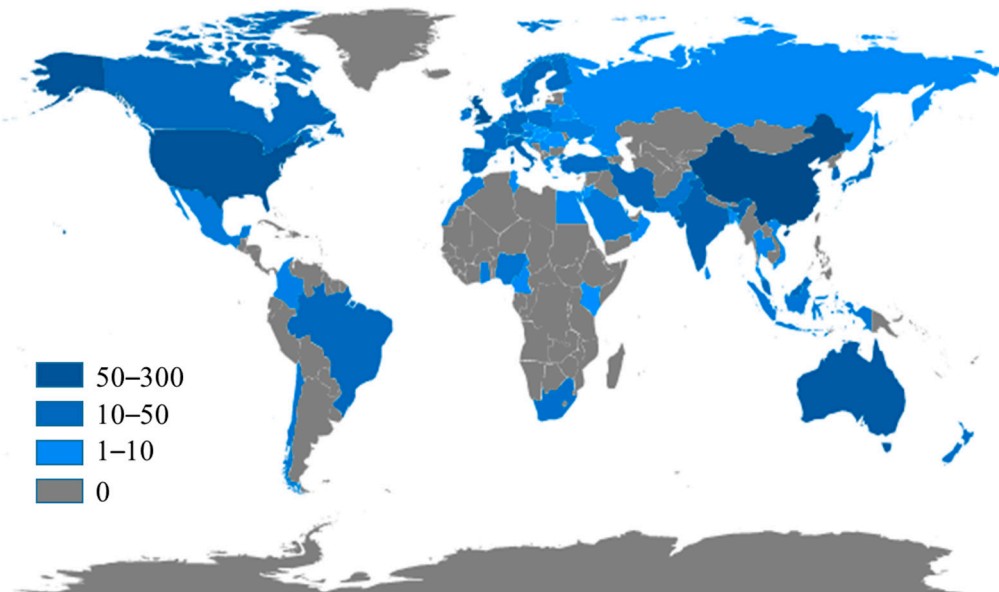

**Figure 4.** Worldwide publications on CSCM.

In order to further analyze the country information in the CSCM literature, the countries/regions of the top 10 corresponding authors are listed in Table 2. Figure 5 shows the time relationship and cooperation relationship of the supply chain management articles of construction projects published by major countries/regions. Table 2 shows that China, the United Kingdom, the United States, and Australia are still leading countries in terms of correspondents, indicating that they are in a leading position in the field of the supply chain management of construction projects. The MCP ratio of all countries is less than 50%, which indicates that transnational cooperation is not sufficient to some extent. In the national cooperation network diagram, if there is an academic cooperation relationship between countries, a connecting line is generated between nodes. The thickness of the line indicates the degree of cooperation (see Figure 5). Figure 5 shows China's outstanding performance in international cooperation. There are 133 papers on international cooperation. Among them, the United States, the United Kingdom, and Australia cooperate frequently, with the frequency of 77, 50, 42, and 11, respectively. In terms of cooperation between countries in the field of CSCM, the strongest cooperation relationship occurs between China and the United States (36 cooperations), followed by China and Australia (23 cooperations). Although the international cooperation network is in its infancy, with the exception of a few countries that maintain close cooperation with foreign countries, the number of domestic

cooperation papers in other countries is more than that with international cooperation, and they mainly focus on independent research.

**Table 2.** Top 10 countries/region of the corresponding authors of CSCM research.

| Country | Articles | Freq | SCP | MCP | MCP_Ratio | Total Citations | Average Article Citations |
|---|---|---|---|---|---|---|---|
| CHINA | 251 | 0.25176 | 179 | 72 | 0.2869 | 2929 | 27.90 |
| UNITED KINGDOM | 135 | 0.13541 | 102 | 33 | 0.2444 | 2645 | 10.54 |
| USA | 105 | 0.10532 | 69 | 36 | 0.3429 | 2595 | 19.22 |
| AUSTRALIA | 67 | 0.0672 | 39 | 28 | 0.4179 | 1334 | 19.91 |
| ITALY | 34 | 0.0341 | 26 | 8 | 0.2353 | 681 | 20.03 |
| INDIA | 31 | 0.03109 | 27 | 4 | 0.129 | 503 | 25.15 |
| IRAN | 26 | 0.02608 | 22 | 4 | 0.1538 | 372 | 14.31 |
| SWEDEN | 24 | 0.02407 | 19 | 5 | 0.2083 | 363 | 15.12 |
| BRAZIL | 20 | 0.02006 | 14 | 6 | 0.3 | 318 | 18.71 |

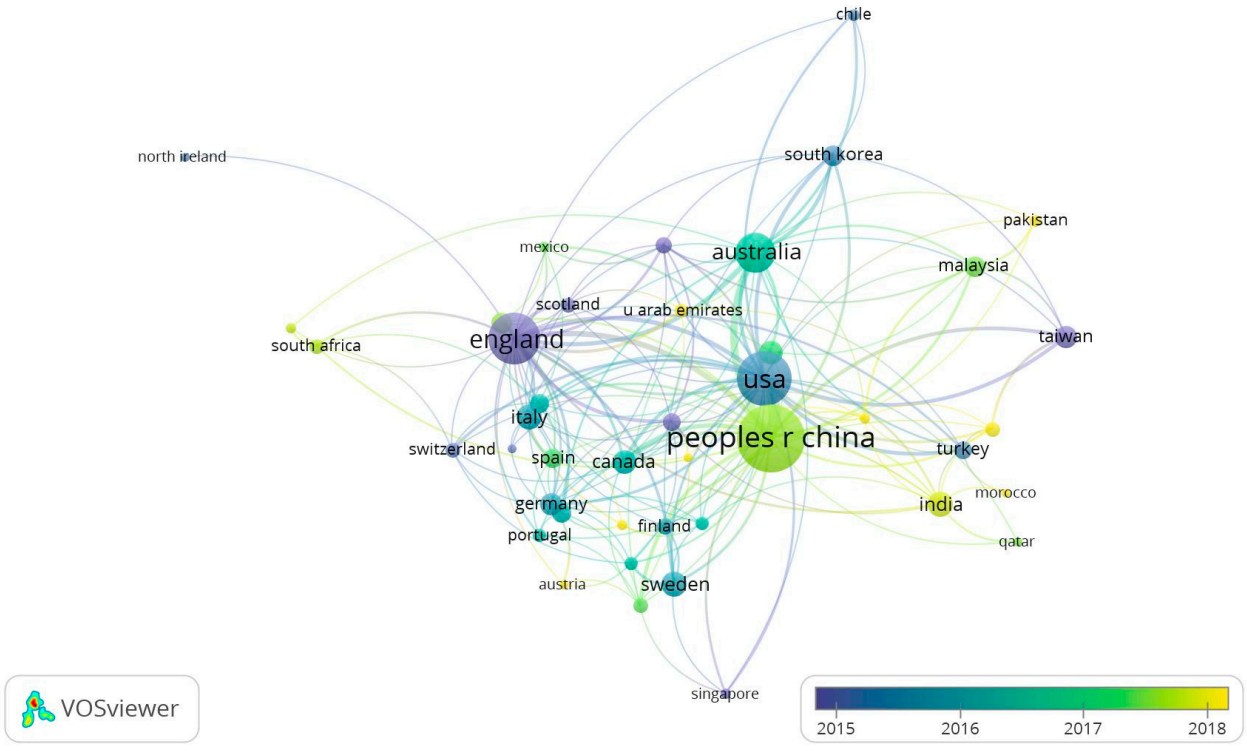

**Figure 5.** Country collaboration network based on co-authorship of CSCM research.

### 3.1.4. The Most Relevant and Influential Journals

The collected documents were analyzed from 332 different journals. This section takes the number of articles and h-indexes as the research indicators to analyze the journals to which the documents belong, so as to identify the most influential journals in the field of CSCM. Figure 6 shows the top 20 journals with the largest number of publications in the field of CSCM. These 20 journals can be regarded as the most recognized journals in the field to some extent. As shown in the figure, the *Journal of Cleaner Production*, *Sustainability*, the *Journal of Construction Engineering and Management,* and *Automation in Construction* are the four largest journals with the largest number of articles published. The main research fields of these journals on construction project supply chain management are green supply chains, construction project supply chains and sustainability, construction project supply chain optimization, and construction project supply chain informatization and simulation. In addition, there are other journals that have published a large num-

ber of articles, including *Supply Chain Management—An International Journal*, *Construction Innovation—England*, *Construction Management and Economics*, and *Engineering Construction and Architectural Management*. The main research fields of these journals are architectural innovation, architectural economy, and construction management. It can be seen that construction project supply chain management is a multidisciplinary research field.

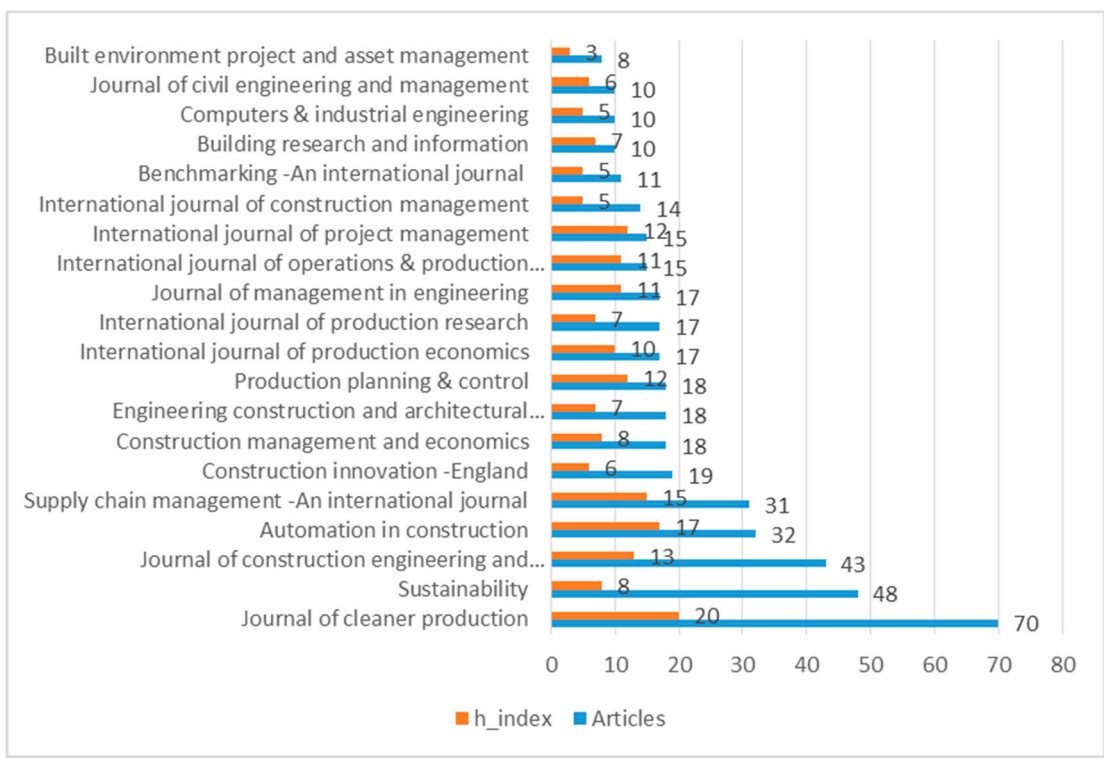

**Figure 6.** Top 20 journals related to CSCM research.

3.1.5. Leading Researchers

The author information contained in the collected data can be used to identify the main authors and countries in the field of CSCM. The h-index, total citations (TCs), number of publications (NPs), and CSCM first publication year (PY-start) of the top 10 influential authors that contributed to CSCM publications are listed in Table 3. Figure 7 exhibits the top 20 authors' publications over time, with the volume of the sphere being proportional to annual NPs and the color depth of the sphere proportional to annual TCs. This paper selected scholars with high citations and assessed their published articles to clearly understand their research scope, since this can be used to identify the research progress in this field and therefore has important research significance. Table 3 indicates that Hosseini has the highest h-index and is the most frequently cited author in this field, with an h-index of 7 and a TC of 154. On the other hand, although Xu has the most published papers, the number of citations is relatively low. The highly cited papers published by Hosseini are mainly concentrated in 2015, 2016, 2018, and 2020 (see Figure 8). The main research direction of Hosseini in the field of CSCM is building model informatization [29], which is followed by the reverse logistics of waste dismantling [30]. Gosling published in a wide range of research topics, including prefabricated housing [31], SC lean management [32], and sustainability [33]. The main research direction of Chileshe and Rameezdeen in the field of CSCM is reverse logistics of demolition waste [34,35]. Rameezdeen [35] emphasized that a reverse logistics supply chain (RLSC), as an environmentally friendly option, has been widely used in the traditional construction SC. Finally, Tatari (and other scholars) have published few articles in the field of CSCM with a high number of citations, suggesting that their publications have a far-reaching impact, thanks to their unique views on sustainable

development over the whole life cycle of construction projects. Tatari [36] quantitatively evaluated the sustainability of the American construction industry from the perspective of the triple bottom line of environment, economy, and society. Tatari [36] proposed that the indirect suppliers in the SC of the construction industry have the strongest impact on sustainable development.

**Table 3.** Top 20 influential authors in the CSCM research field.

| Author | h_index | TC | NP | PY_start |
|---|---|---|---|---|
| Xu JP | 4 | 35 | 9 | 2013 |
| Chileshe N | 7 | 151 | 8 | 2015 |
| Hosseini MR | 7 | 154 | 8 | 2015 |
| Edwards DJ | 4 | 100 | 7 | 2004 |
| Gosling J | 6 | 108 | 7 | 2013 |
| Li Y | 2 | 29 | 7 | 2018 |
| Rameezdeen R | 6 | 141 | 7 | 2015 |
| Shen GQ | 4 | 129 | 7 | 2016 |
| Wang XY | 5 | 114 | 7 | 2016 |
| Kucukvar M | 6 | 227 | 6 | 2012 |
| Loosemore M | 6 | 84 | 6 | 2016 |
| Zhang Y | 3 | 27 | 6 | 2011 |
| Aziz Z | 4 | 51 | 5 | 2017 |
| Haas CT | 3 | 72 | 5 | 2010 |
| Li H | 2 | 105 | 5 | 2011 |
| Lim BTH | 5 | 96 | 5 | 2011 |
| Meng XH | 4 | 254 | 5 | 2011 |
| Moon S | 4 | 37 | 5 | 2015 |
| Naim M | 5 | 87 | 5 | 2003 |
| Tatari O | 5 | 213 | 5 | 2012 |

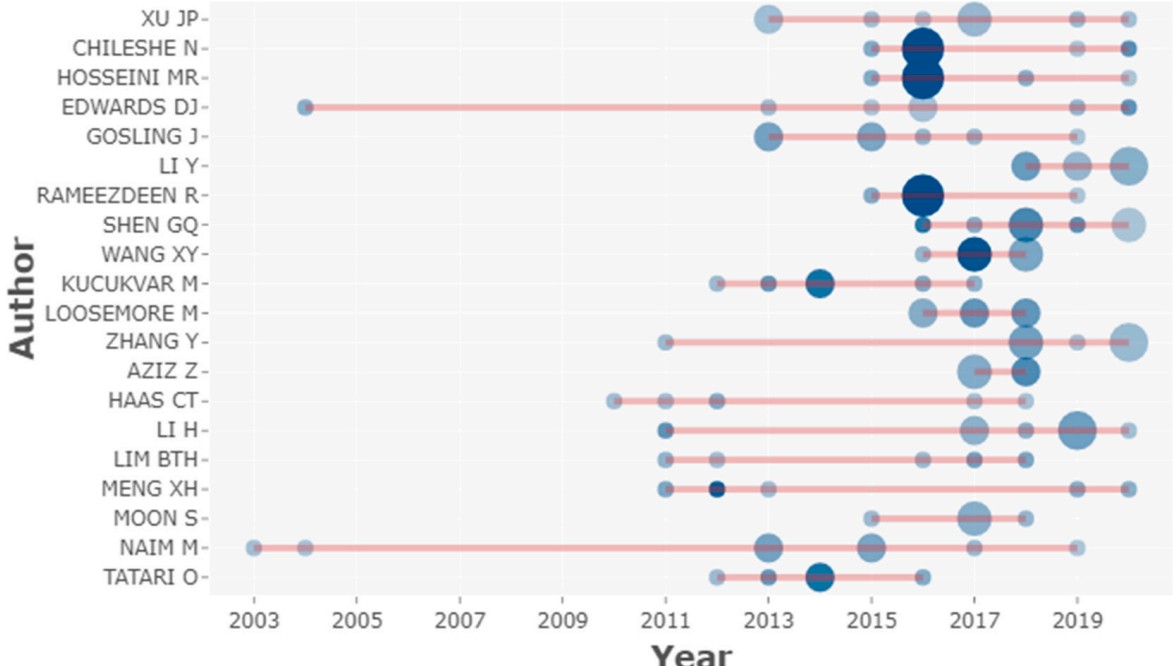

**Figure 7.** Top 20 authors' publications over time in the CSCM research field.

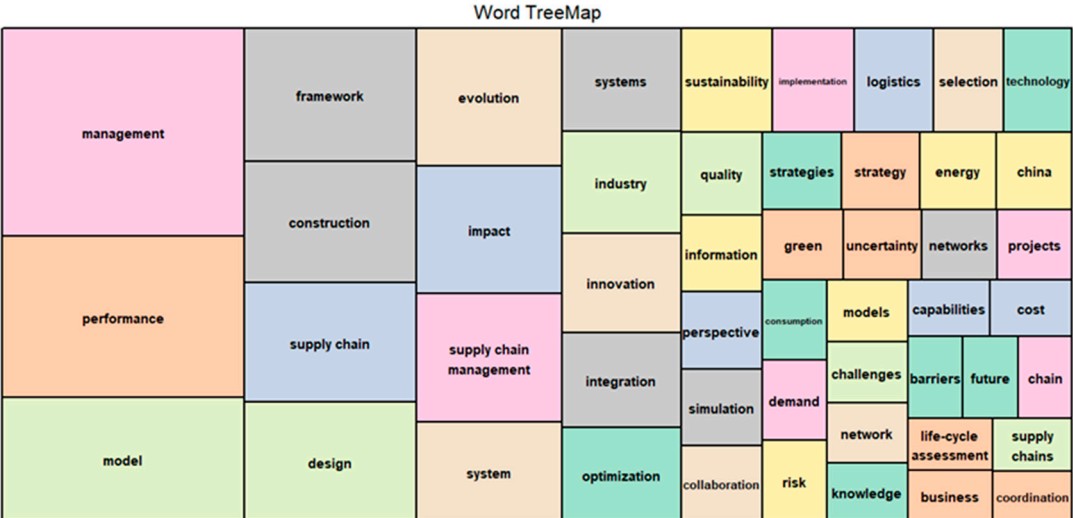

**Figure 8.** CSCM's keyword clouds.

### 3.2. Keyword Analysis

3.2.1. Analysis of Keywords

At the beginning of the analysis, the frequency of keywords was statistically sorted, and the frequency of keywords preliminarily determined the research hotspots. Keywords analysis can be used to identify hot topics, since keywords are used by the authors as a clear, representative, and concise descriptions of the research content [37]. First, Bibliometrix was used to obtain information about the frequency of keywords in the CSCM domain. As shown in Figure 9 and Table 4, keyword clouds were generated using a high-frequency keyword list identified in the corpus. The size of keywords was positively correlated with the frequency of their occurrence in the dataset. The resulting word cloud map represents a clear and complete graphic display of hot topics in the field of CSCM research, indicating management, performance, model, framework, construction, supply chain, design, evolution, impact, and system as the top 10 keywords in the CSCM domain.

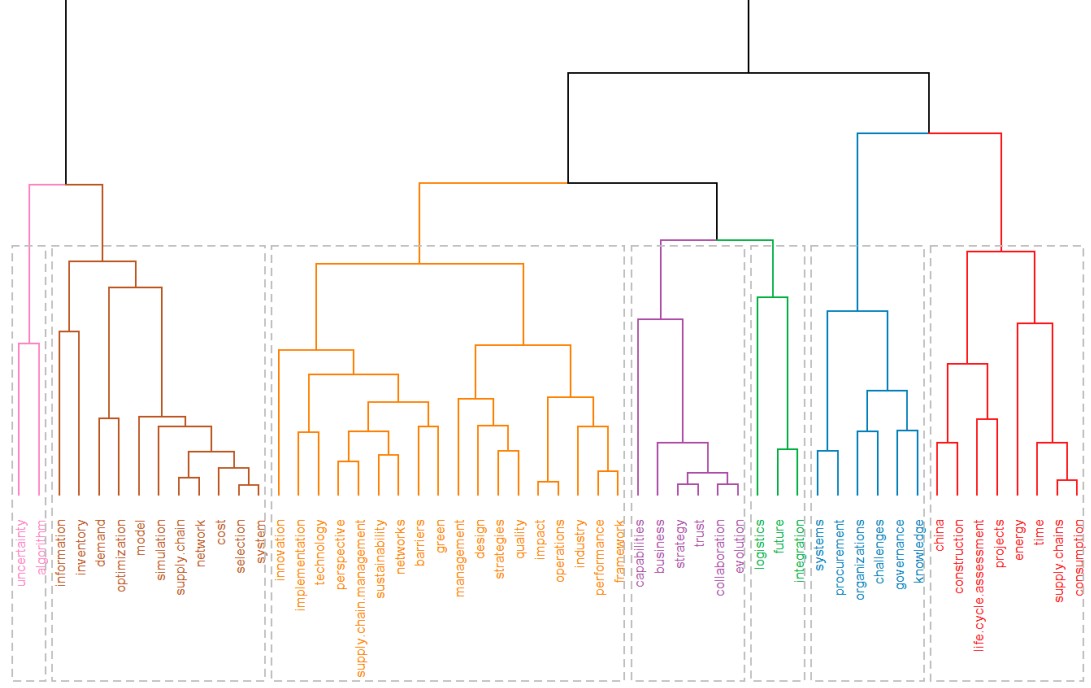

**Figure 9.** Hierarchical clustering of keywords in CSCM domain dendrogram.

**Table 4.** High-frequency keywords and their frequency in the field of CSCM.

| Keywords | Frequency | Keywords | Frequency |
|---|---|---|---|
| management | 468 | integration | 105 |
| performance | 363 | optimization | 103 |
| model | 278 | sustainability | 88 |
| framework | 211 | implementation | 79 |
| construction | 194 | logistics | 76 |
| supply chain | 190 | selection | 67 |
| design | 189 | technology | 66 |
| evolution | 186 | quality | 63 |
| impact | 174 | information | 58 |
| supply chain management | 174 | perspective | 58 |
| system | 134 | simulation | 58 |
| industry | 112 | collaboration | 57 |
| innovation | 109 | strategies | 57 |

Then, the CSCM conceptual structure diagram was further generated. In the conceptual structure diagram, the hierarchical clustering method was used to analyze the frequency of the simultaneous occurrence of two keywords through statistical methods, and the keyword network relationship was simplified into several relatively small groups [38]. The steps of clustering were as follows: firstly, the frequency of each keyword appearing at the same time was counted, and the keywords with high frequency were regarded as a class. Then, the two clustering keywords were combined with the highest similarity to form a new big cluster. Then, the new cluster was merged with the cluster with the highest similarity, and the merge was repeated in this way until all individuals were combined together. Finally, the whole classification system is presented in the form of a tree diagram (Figure 9). On the basis of cluster analysis, this paper used multiple correspondence analysis (MCA) to further analyze the categories of keywords. MCA is a commonly used sociological method. It uses plane distance to reflect the similarity between keywords, which can further clarify the difference of importance between keywords based on cluster analysis. The closer to the edge, the narrower the research topic, or the stronger the transition to another topic [39] (see Figure 10).

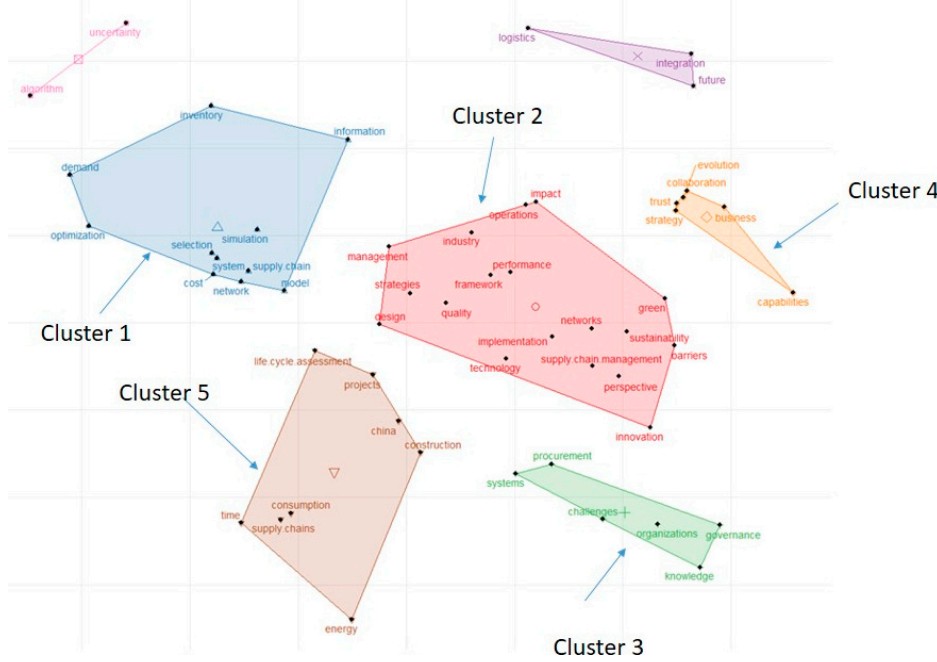

**Figure 10.** Conceptual structure map of the CSCM research field.

Cluster 1 is composed of 11 keywords, which highlights the research goal of CSCM. Among these keywords, "algorithm optimization", "model simulation", "network optimization ", "system", and "cost optimization" are keywords to describe the main SC research contents for construction projects. Popular methods and technologies used include material requirements planning (MRP), enterprise resource planning (ERP), building information modeling (BIM), and optimization technology (OT) [40–42]. A variety of methods and techniques (e.g., inventory management, project planning, and control) were developed to improve construction performance for material control, on-site transportation management, and project planning (Wang [41], Min [43]). Voordijk [44] proposed a physical non-matching system model including elements such as inventory, transportation, loading and unloading, as well as warehousing, to cost the thermal insulation materials supply chain. Similarly, a simulation toolkit, which is compatible with other construction simulation tools, was developed to model different SC problems [45]. Using this toolkit, a simulation model of the impact of SC problems on the productivity of a real construction project could be established.

Cluster 2 is the largest cluster with 18 keywords. Clustering of these keywords showed that the topics around specific keywords are often studied in the same paper. Unlike cluster 1, "network" is the subject of CSCM research methods in this cluster. This further illustrates the popularity of the keyword "network" in the field of CSCM. Another common focus is on sustainable and green SCM. Keywords such as "green" and "sustainability" describe a unique theme in the field of CSCM. The development of sustainable green supply chain management (GSCM) is complex and requires collaboration and integration with different activities of SC partners. For example, Lloyd [46] assessed the purchase scale and frequency of market participation, the basis of product selection, the purchase impact, and price sensitivity to identify products and market methods that may increase the success of certified wood products. Lloyd [46] also provided suggestions for expanding the market for green building materials. Based on an understanding of the whole CSC and its impact on sustainable construction, Ravetz [47] discussed a series of indicators and benchmarks for sustainable construction, considering both global climate and resources.

Cluster 3 is mainly related to the research object of CSCM. "Procurement", "systems", "organization", "governance", and "knowledge" are representative keywords of this cluster. In recent years, the coordination and integration of various subjects in CSCM has become the focus of researchers. In the ever-changing and highly competitive market environment, the key to the success of an enterprise largely depends on whether the enterprise can quickly respond to changes in the business environment and create value for the market it serves. To achieve this, enterprises need to establish close business partnerships with suppliers, customers, and even competitors. Being a member of a certain SC, firms should thus share information with other chain members and coordinate the planning [48]; provide fast, flexible, and efficient support and services to its end customers; as well as set up new business operation models. Establishing an integrated cross-enterprise support system is critical in achieving this marketing goal [49], because of various independent information systems within an enterprise [50]. Moreover, to better respond to changes in the business environment and gain the initiative in fierce market competition, enterprises need to conduct system integration with business partners to provide better services for their end customers [51].

Cluster 4 is an extension of the research object in theme 3, but this theme focuses on the optimization of supply chain partnership. "Trust", "collaboration", and "capabilities" are representative keywords of this cluster. Enterprises from different countries and regions that participate in strategic alliances inevitably experience the coexistence of various cultures. A lack of goal consistency, commitment, and trust between manufacturers and suppliers is considered as a major obstacle for management integration in the supplier relationship [8]. Integrated network partners should thus adopt a team-oriented approach to maximize mutual benefit, which would encourage suppliers to align and achieve their goals with those of manufacturers. Trust is a complex and subtle phenomenon that en-

ables cooperative strategies and relationship advantages [52]. Investing in integrity not only changes the company's reputation for integrity but also its ability to participate in value co-creation programs [53]. Information sharing in SCs is also based on the stability and trust of the SC relationship. In the existing literature, scholars almost unanimously advocate information sharing. Companies need to build information platforms for sharing information among relevant parties to make decisions together in an effective manner.

Cluster 5 highlights the limited conditions of CSCM research. For example, "life cycle assessment" is the limitation of the research time range, "China" is the limitation of the research area, and "time" and "energy" is the limitation of the research object. It seems life cycle assessment is one of the most common scenarios in supply chain management applications [54]. As one of the fastest growing countries in the past 10 years, China is one of the most key countries in the study of CSCM [50].

In addition to these significant clusters, the rest of the sets (groups 6–7) are small clusters of conceptual diagrams representing other high-frequency keywords in the CSCM field. For example, "uncertain" in cluster 7 is a frequent concern in the area of CSCM. "Logistics" in cluster 6 is often studied in CSCM research.

A co-occurrence network of the first 50 high-frequency keywords with time information is established in Figure 11. In the co-occurrence network, the overall distance between keywords reflects their correlation. In general, a shorter distance between two keywords implies a stronger relationship between them [55]. The color of each keyword represents its average publication year, which is determined by averaging the publication year of all documents with this keyword in the title or abstract. Keywords that are used more frequently in 2014 are shown in blue, while those that are used more frequently in 2018 are shown in yellow. In terms of the time of their co-occurrence of the network, the average release year of these high-frequency keywords mainly ranges between 2015 and 2017. Most studies around 2016 focused on content related to improving performance ("blue" keywords), such as "project management", "risk management", "RFID", "cooperation", and "trust". Keywords in yellow, "sustainability", "circular economy", "waste management", "case studies", "uncertainty", and "social responsibility", seem to have recently attracted increasing attention, since the average publication year is close to 2018. These keywords are more likely to continue to flourish in the CSCM field.

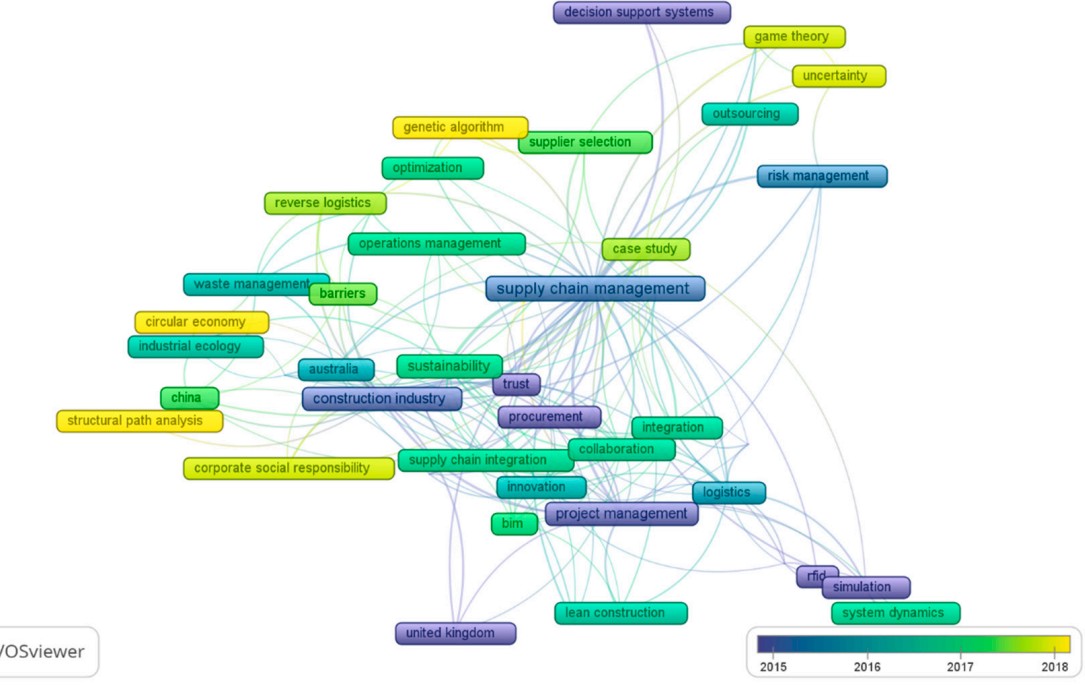

**Figure 11.** Co-occurrence network of high-frequency keywords of the CSCM literature.

### 3.2.2. Analysis of the Evolution and Development of Keywords

Based on the complete analysis of keywords' current situation, this section attempts to analyze the theme evolution of CSCM research from 1998 to 2021 from a dynamic perspective. Drawing upon lessons from some existing papers to detect change in topics [23,56], this paper divided the research period (1998–2021) into four consecutive sub-periods, considering that the first period was fixed at 16 years due to the limited number of articles published in the early stage. Compared with subsequent periods, this situation ensured that the size of the first sub-period was reasonable. The last three periods were determined to cover two years each. Therefore, the whole study period (1998–2021) was divided into four consecutive periods: 1998–2014, 2015–2017, 2018–2019, and 2020–2021.

A strategic map of CSCM research for each sub-period was built with Bibliometrix using co-word analysis based on author keywords, as displayed in Figure 12. The sizes of the spheres in Figure 11 are proportional to the keyword frequency of each topic highlighted in each individual sphere. Keywords that appeared at least twice were retained, while different forms of keywords reflecting a common theme (e.g., SCM and CSCM) were removed. Highly relevant keywords were then grouped into topic clusters, which were named after the most frequent keywords. Each sub-graph in Figure 12 is divided into four quadrants, representing different topics. Two metrics, including centrality and density, are used to characterize each topic cluster. Figure 12 shows that the topics with a large number of publications are mainly located in the fourth quadrant, indicating that basic themes and horizontal themes are the main priorities in the field of CSCM. As can be seen from the picture, in the field of early CSCM, more developed countries represented by the UK have explored the optimization of management, but they have also begun to study the effective use of resources and environmental protection. After the accumulation of research, scholars at this stage are more focused on exploring the research on efficiency in the fast-developing countries represented by China. Scholars are also increasingly interested in case-interest studies, such as the study of Aswan High Dam.

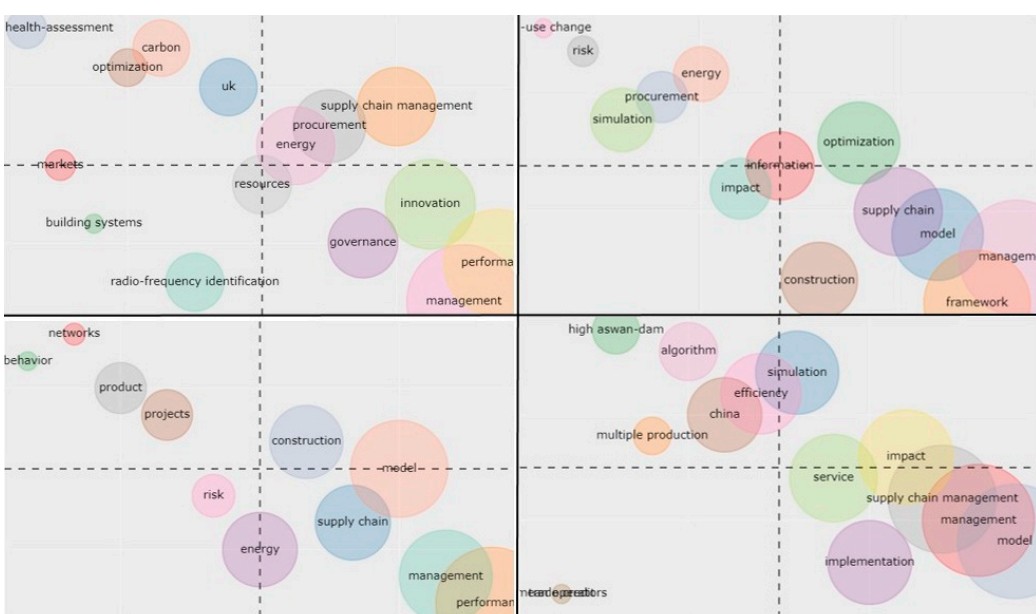

**Figure 12.** Strategic diagrams of CSCM research (1998–2021).

Furthermore, a Sankey diagram was constructed to analyze how topics cluster and to detect the main evolution paths of these topics, as shown in Figure 13. Each node represents a topic cluster marked by keywords with the highest frequency and corresponding sub-periods. The size of the nodes is proportional to the number of keywords associated with the corresponding topic. The flow between the nodes indicates the evolution direction of the topic cluster. The bandwidth is proportional to the inclusion index between two

linked topics. As seen in Figure 13, the number of links between topics increases over time. Specific topics, appearing in the far right hand side, have developed steadily, showing their increasing importance. These main topics discussed in the corpus are: (1) SC integration and SCM, (2) SC process design and optimization, (3) the application of advanced technology, and (4) SC sustainable management.

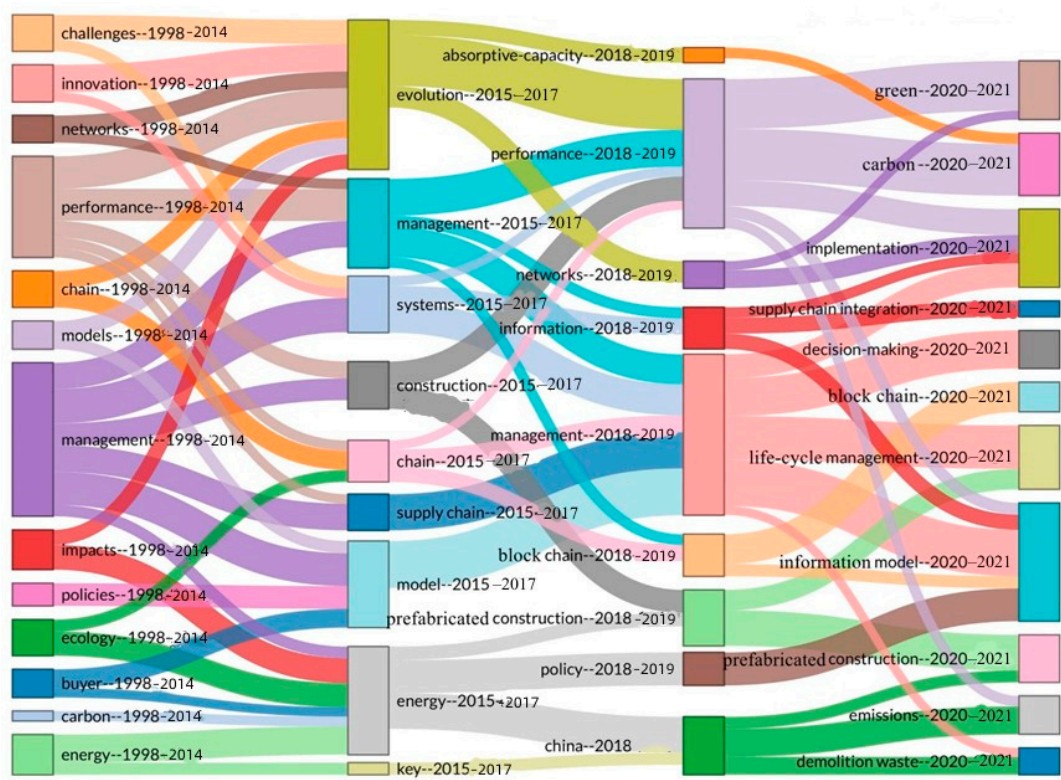

**Figure 13.** Thematic evolution of CSCM research (1998–2021).

The development path of SC integration and management can be described as concrete→procedural justice→trust; management→collaboration; management→ institutional pressures→implementation; performance→collaboration→ performance; dominance constrains→culture→performance; trust→management→ performance. This topic is the interpretation of the research objectives in CSCM, from early attention to the impact of "concrete", "management", and "dominance constrains" in the supply chain on construction projects, to the improvement of soft power such as "culture", "trust" and "collaboration" on the performance of construction projects. SC integration and management include both external and internal integration. The external integration of SCM ensures the real-time exchange of information by searching for an effective inter-organizational cooperation model. This realizes the sharing of market demand, inventory status, production planning, demand forecasting, and delivery planning [57]. The advantages of external integration have been widely reported in the manufacturing and logistics industry. The sustainable development of an economy can be promoted by enhancing enterprises cooperation.

The development path of SC process design and optimization can be described as life cycle assessment→framework→management; optimization→operations; concrete→procedural justice→trust; the CSC is very different from the traditional SC, because the supply needs to be continuously adjusted according to the different time and task requirements of the contractor in the construction project [58] and because of the complexity of the construction project and the huge construction cost of the long-term supply chain system in the early stage. In order to optimize the cost of using a supply chain in each stage of a construction project, scholars mainly use simulation to simulate its effectiveness in process design. In the research of supply chain effectiveness optimization, supply chain process optimization

with time and cost as the goal is developing towards multi-objective optimization. Among them, the genetic algorithm, ant colony optimization, and particle swarm optimization are commonly used research methods [59]. Similarly, in the study of supply chain effectiveness optimization, the single-objective optimization of time, quality, and cost has been transformed into the multi-objective comprehensive optimization of time, quality, and cost [17].

The application of advanced technology mentioned in the corpus are radio frequency identification→ model→model; model→ model→ management; life cycle assessment→management→BIM; framework→ BIM→ construction. The application of advanced technology is an attempt by the academic community to effectively improve "management" and "performance". In the whole process of the application of CSCM, technology application focuses on two aspects: the application of management technology and the application of digital technology. Management technology focuses on the rapid communication of all stakeholders in the supply chain system, such as mobile-terminal-enabling technology (RFID and GPS), virtual reality mobile clients, and application services (AR/VR and cloud computing) [50]. The application of digital technology focuses on efficiency improvement in the supply chain system, for example, mobile terminal equipment (PDA, mobile phones, and wearable devices) and prefabricated component management systems based on BIM and RFID [60].

Sustainable management can be described as: sustainability, construction, management; model, governance, implementation; consumption, SC, environment. The sustainability of SCM focuses on construction waste disposal and the management of externalities. The weight of topic clusters in the evolution path follows an increasing trend, indicating that the waste disposal and externality management of CSCM is attracting increasing attention. This trend is likely to continue and grow into a paramount field in CSCM research. Sustainable performance evaluation mainly includes environmental performance, economic performance, and social performance [61]. In terms of economic benefits, the implementation of effective SCM for construction projects has been shown to hold the potential to increase enterprise profits by nearly 40% [62]. The effective integration of CSCM offers greater social benefits, such as reducing labor consumption and saving energy [63].

## 4. Discussion

The concept of the supply chain was introduced from manufacturing to construction [1]. One of the important purposes of scholars' research on supply chain management is to obtain sustainable advantages of enterprises in supply chain competition through more comprehensive supply chain management. Therefore, the research theme of CSCM was differentiated from "performance" and "management" in the early stage. The types of topics have been sub-divided with the gradual enrichment of research. From the analysis of the evolution of the key words in the CSCM topic evolution, it can be seen that the research topic is easier to shift to a figurative theme (collaboration, BIM and evolution), and easier to shift from a broader theme (performance and management). In addition, some research topics are important bridges and mediators in the evolution of topics and are key links in research development and change. For example, framework is the key node in the evolution of life cycle assessment to BIM research. The theme of institutional pressure also provides a theoretical transition for the research of implementation and management. Under the guidance of the management concept of maximizing overall benefits, enterprises optimize supply chain processes by applying advanced technologies (such as radio-frequency identification and BIM) to achieve the goal of performance improvement. In general, management optimization and performance improvement are the ultimate goals of supply chain management of construction projects. Only with the joint development of the two can CSCM be efficient and sustainable. Therefore, this paper divided the research of CSCM according to the management level and performance level and divided it into four theme directions to discuss future development in combination with the content of the theme differentiation of keyword evolution and development (Figure 14).

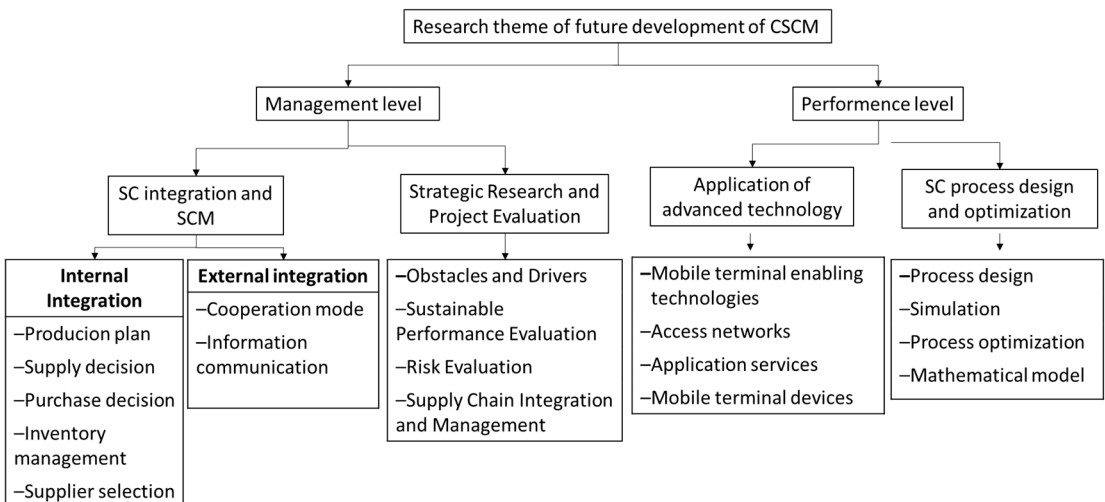

**Figure 14.** Future development theme direction.

### 4.1. Supply Chain Integration

Although most managers have realized the importance of supply chain management in performance improvement, in the practical application of CSCM, only the on-site construction link is paid enough attention, and the concept of CSCM has not been applied to whole-process management [64]. Therefore, for CSCM, the whole process from the on-site construction environment to the construction project is an important research topic in the future. Therefore, the following two aspects need more attention: the obstacles to cooperation within the supply chain and the cooperation and communication of integration outside the supply chain. For the supply chain itself, first of all, cost is the main obstacle to the use of CSC. If the use of a functional supply chain leads to increased cost, managers will tend to use other methods. For example, the supply chain decision-making time is too long due to poor communication, resulting in increased costs [65]. Secondly, another important obstacle is the complexity of the implementation process [66]. For example, for suppliers, it is necessary to develop extremely complex inventory management plans for the needs of multiple contractors [67]. Compared with internal cooperation, external integration focuses on the supply chain cooperation mode and information communication. The main purpose of the research on the cooperation mode is to design a reasonable cost-sharing mode and risk and benefit allocation [68]. Game theory is the most common method used to study cooperation and competition in a supply chain [17]. The external integration of the supply chain ensures the smooth transmission of information throughout the supply chain, so as to better help participants achieve their interests and promote the use and promotion of the supply chain.

### 4.2. Supply Chain Strategic Research

The study of supply chain strategies is very important in the study of supply chains. In the course of CSCM research, with the development of research, the focus of research has been constantly changing, from the early 'supplier selection', 'decision support system', and 'risk management' to 'circular economy', 'corporate social responsibility', and 'reverse logistics'. As the concept of CSCM continues to spread, the hot research areas have also transited from the United Kingdom in the early stages to Australia and finally to China. CSCM researchers have changed from focusing only on the efficiency and competitiveness of individual individuals in the supply chain in the early stage to focusing more on the improvement and sustainability of the overall efficiency in the supply chain. In the first decade of the 21st century, researchers of supply chain management saw that the focus of research was shifting from supply chain management to strategic supply chain management. Melnyk et al. (2006) conducted a Delphi study (together with leading academic researchers and supply chain practitioners) to determine the most critical strategic issues faced by

supply chain managers in 2010 and beyond. The research results show that these five issues are the most important: supply chain disruption and supply chain risk, leadership within the supply chain, the management of the timely delivery of goods and services, the management of product innovation by using supply chain capabilities, and the seamless exchange of information within the supply chain by using appropriate technologies. After ten years of development, as well as the increasing requirements of construction projects on economic, social and environmental aspects, this study summarizes the four measures needed to improve the existing capacity and meet the future supply chain demand in future research through the status quo and evolution analysis results of key words:

I     Using supply chain optimization models, including optimization, risk, and cost.
II    Applying corresponding technology to manage the whole process of a supply chain.
III   The comprehensive management of the economic, social, and environmental aspects of the construction project supply chain.
IV    Managing using process orientation with appropriate measures.

### 4.3. Supply Chain Process Design and Optimization

Optimization framework appears in the hot keywords of CSCM, reflecting the interest of CSCM researchers in process design and optimization. The implementation process of construction projects needs to complete multiple objectives at the same time, such as the requirements of time limit, quality, and cost. In the research of CSCM, the research objects of researchers are also gradually diversified, and the optimization objectives are transferred from traditional single-objective optimization (such as duration) to multi-objective optimization. Among the optimization objectives, construction progress is considered as the primary objective. During the construction process, the delay of material mobilization may lead to the extension of the construction period and the increase in the labor cost. However, early entry will also increase storage costs and waste space [69]. Researchers and managers need to integrate the cost objectives of the construction project, resource constraints, workers, material storage, and workspace into the production scheduling model to establish the production scheduling scheme with the lowest production cost [70]. Supply chain process design and optimization is the best way to improve the production efficiency and competitiveness of assembly, construction, suppliers and contractors, and achieve cost control. It is worth noting that most of the current research focuses on the early stage of the whole life cycle of the construction project [17], such as the optimization of the design and construction stages. The end demolition stage of the construction project is ignored. In the keyword analysis, the appearance of "life-cycle assessment" means that researchers pay attention to the whole life cycle of the construction project. Therefore, in future research, researchers should carry out more research on the end of the life cycle.

### 4.4. Application of Advanced Technology

In the keyword analysis, lean construction, BIM, and RFID reflect the researchers' discussion on the application of advanced technology. Among them, it can be divided into management technology and digital technology according to different research objects. Management technology mainly includes lean construction and agile construction. Lean construction is an improvement on traditional construction management [71]. Lean construction can be applied in the design and procurement stages to continuously improve the process and improve efficiency to meet the changing needs of different demanders. However, the implementation of lean construction needs the coordination of the whole supply chain system, so the success of lean depends on the commitment of stakeholders in the supply chain system to continuous improvement [25,72]. Because the nature of the construction industry is different from other manufacturing industries, the application of lean in CSC projects is also different. Lean construction can be used to strengthen processes and eliminate waste and mistakes on construction sites. Lean construction can also make a working environment clean, safe, and efficient. The agile principle is an important supplement to the lean concept in CSCM to reduce the uncertainty caused by time in the

lean construction process [73]. The effective combination of management technology and information technology is another trend in the application of CSCM. For example, the integration of lean and BIM is considered to be an effective way to improve the performance of the construction phase and improve the efficiency of construction throughout the life cycle of the construction project [72]. Although some people have proposed to integrate BIM and lean methods into the construction industry, no research has been found to optimize construction management by combining supply chain management (SCM), logistics management (BIM), and lean (lean). The integration of the three concepts of the construction industry (SCM, BIM, and lean) should not be limited to construction management but should be extended to all stages of the whole life cycle of construction projects. This limitation provides an opportunity for further research on construction logistics and supply chain management based on BIM.

### 5. Managerial Insights

In this section, we propose management opinions based on the results of bibliometric analysis and the discussion in Section 4. This paper believes that sustainability and lean construction are the most important in the field of CSCM. Therefore, this paper attempts to analyze the methods of managing CSC from the following two points.

At present, it is generally recognized that there are three bottom lines (environment, society, and economy [36]) in the sustainability research of CSCM. However, with the continuous development of research, more indicators have been included in sustainability research, including environment, society, economy, population, culture, and politics. In future research, how to coordinate the interaction between sustainable factors under the premise of paying attention to the overall objectives of time, quality, and cost is the focus and difficulty of future CSCM research. In the study of sustainability, scholars believe that natural resources have a high binding force on economic growth from a long-term perspective. The value of natural capital is not only reflected in the economic value of resources but also in its unique ecosystem service value and the opportunity cost of consuming natural capital [74]. Therefore, we should strengthen the basic theory research of sustainable utilization in the construction project supply chain based on the science of sustainable development. In sustainable exploration, waste reverse supply chain recycling and green supply chain management will be important development directions in the future.

Finally, the concept of lean construction in the supply chain management process of construction projects is worth promoting. Lean construction aims to continuously improve the process in each process of the construction project in the face of changing material demand, so as to reduce waste and improve the efficiency of stakeholders in the system. In the implementation process of lean construction, it is more a kind of ideological dissemination than the application of lean methods. Therefore, the effective promotion of the concept of lean construction depends on the firm determination of every stakeholder in the CSC to continuously improve efficiency [25,72]. Specifically, promoting the concept of lean construction requires cooperation among stakeholders. In the planning and design stage, all stakeholders will be included in the system framework for consideration. Because there are many stakeholders in a construction project and the operation is complex, lean construction needs to formulate the specific objectives, standards and performance indicators of all stakeholders in the construction process at the planning and design stages during the implementation process, so as to provide the optimal value for the ultimate achievement of the owner's objectives.

### 6. Conclusions

This paper introduces the general situation of CSCM research through econometric analysis using the newly developed Bibliometrix tool in R. Moreover, the knowledge structure of this field is explored through content analysis. This analysis is based on a rich, reliable, and high-quality data set, including 2282 journal articles published between 1998 and 2021. These publications represent an overview of the research on CSCM that intro-

duces the annual quantity distribution of the CSCM literature, citations, the most relevant and influential journals, as well as leading researchers and countries. The main findings in this area are summarized as follows: (1) CSCM research is divided into three stages according to 2007 and 2014. Since 2007, CSCM has become the research focus of relevant scholars, and the number of published articles has also increased rapidly. On the whole, the average number of citations increased but fluctuated to a certain extent in 2009, 2012, and 2014, which represent three peaks. These results show that over time, the global attention to CSCM research has gradually increased, which is related to the increasingly complex engineering construction in the world in recent years. (2) The development of citations indicates that the best paper in the field of CSCM was published by Briscoe [25] in *Supply Chain Management: An International Journal.* Efforts such as those by Irizarry [28] to apply new technologies to CSCM have also made important contributions to the development of CSCM. (3) Hosseini MR is the most represented researcher in this field, and his research direction is building model informatization. This was followed by Gosling J, Rameezdeen R, and Kucukvar M. Related research directions are the reverse logistics of demolishing waste, prefabricated housing, and SC lean management.

With regard to the knowledge structure of the CSCM domain, keywords were analyzed to determine the core elements of the knowledge base within this domain. Based on keyword analysis, this paper introduced the relevant keywords cloud, the top 10 keywords with frequency changes, a conceptual structure map, network co-occurrence with time information, a strategy map, and a Sanji diagrid. These methods were used to identify the main semantic topics hidden in the text data. From the perspective of the current situation and historical dynamic evolution, the evolution of the theme can be identified in the field of CSCM. The main results are summarized as follows: (1) In the analysis of high-frequency keywords in the field of CSCM, it is found that the top 10 keywords (according to their frequency) are *management, performance, model, framework, construction, supply chain, design, evolution, impact, and system.* (2) According to the results of the cluster analysis of these high-frequency keywords in the CSCM field, the four main research hotspots include i. SC simulation research, ii. sustainable and green SCM, iii. SC overall collaboration and SC integration, and iii. upstream and downstream relationships as well as trust and communication. (3) From the perspective of the timing of co-occurrence networks, the keywords of "sustainability", "circular economy", "waste management", "case studies", "uncertainty", and "social responsibility" are more likely to attract more attention in the future (see Figure 10). (4) By analyzing the evolution of themes, several main theme areas have been identified: (1) SC integration and SCM, (2) SC process design and optimization, (3) the application of advanced technologies, and (4) sustainable SCM.

**Author Contributions:** S.W. conceived this study. H.T. designed and completed the paper in English. F.Y. provided significant advice. G.W. revised the manuscript. All authors have read and agreed to the published version of the manuscript.

**Funding:** This study is supported by the National Natural Science Foundation of China (71972018).

**Institutional Review Board Statement:** Not applicable.

**Informed Consent Statement:** Informed consent was obtained from all subjects involved in the study.

**Data Availability Statement:** Not applicable.

**Conflicts of Interest:** The authors declare no conflict of interest.

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
