# Peer review of "Exploring the Global Research Trends of Supply Chain Management of Construction Projects Based on a Bibliometric Analysis: Current Status and Future Prospects"

_buildings, doi:10.3390/buildings13020373_

Round 1
Reviewer 1 Report
In my opinion, this paper can be accepted after consideration to below comments:
1. Litruture review is very cheap and you should improve the quality of study history. What is the importance of your study?
2. You should use both citescore and VOSviewer (Country distribution and researchers) and compare their outcomes with Bibliometrix.
3. You should mention to classification and clustering algorithm of Bibliometrix.
4. What foresight result you can obtain after evaluation of trend? Discuss it in the main passage.
5. Add circular economy concept in your bibliometrix analysis. You should discuss CE more.
6. As well supply chain, you should insert a topic in the field of Closed-Loop Supply Chain.
Your paper has not any novelty and your disussions can be assumed as your innoviations. Therefore, in reconsideration process, I will seek your discussions.
Author Response
请参阅附件

Reviewer 2 Report
The paper faces the supply chain management of construction projects through the investigation of the global research trends using bibliometric analysis. The research contents are clear and well organised, as are the future developments and analysis perspectives. The bibliographic references are representative and mostly recent. However, the following changes are suggested:
1) Correct, where necessary, references to bibliographical sources in the body of the text (e.g. lines 45, 47, etc.);
2) Revise the numbering of the sub-sections. In particular, sub-section “2.1 Methods” (line 99) has the same numbering as the next sub-section (“Data sources”, line 112).
3) I would suggest to write, in sub-section “3.1 Overview of construction supply chain management research” (line 132), an introduction on what is going to be explored/analysed in the subsequent sub-sections.
It would be useful to clarify if the analysed sector literature refers to the period 1998-2022 (as indicated in the abstract in line 15), or to the period 1998-2020 (as indicated in line 81), or to the period 1998-2021 (as indicated in line 353).
Round 2
Reviewer 1 Report
In my opinion, it can be accepted.
Author Response
Dear reviewer:
Thank you for your careful review. We really appreciate your efforts in reviewing our manuscript during this unprecedented and challenging time. We wish good health to you, your family, and community. Your careful review has helped to make our study clearer and more comprehensive.
Handong
2023.1.16